# Health economic evaluation of noninvasive prenatal testing and serum screening for down syndrome

**Gefei Xiao** * , **Yanling Zhao, Wuyan Huang, Liqing Hu, Guoqing Wang, Huayu Luo**

Department of Clinic Laboratory (Institute of medical genetics), Zhuhai Center for Maternal and Child Healthcare, Zhuhai, Guangdong province, China

* xgf8111_cn@hotmail.com

## Abstract

### Background

Down syndrome (DS), also known as trisomy 21 (T21), is the most common genetic disorder associated with intellectual disability. There are two methods commonly used for prenatal testing of DS: serum screening (SS) for biomarkers in maternal serum and noninvasive prenatal testing (NIPT) for aneuploidy by cell-free DNA (cfDNA) in maternal plasma. However, cost-effectiveness analyses of these two methods are mostly based on data derived from simulations with various models, with theoretical values calculated. In this study, we statistically analyzed clinical DS screening data and pregnancy outcomes during the follow-up of pregnant women in Zhuhai City, China. The economics of the two mainstream prenatal DS screening methods was evaluated from a public health perspective.

### Methods

A retrospective analysis was performed on the data of 17,363 pregnant women who received SS and NIPT during gestation in Zhuhai from 2018 to 2019, and a cost-effectiveness analysis was performed with four screening strategies. In strategy I, all pregnant women received SS, and those with T21 risk ≥1/270 had invasive prenatal diagnosis (IPD). In strategy II, all pregnant women received SS, those with T21 risk ≥ 1/270 had IPD, and those with 1/270 > T21 risk ≥ 1/1,000 had NIPT; then, women at high risk based on NIPT also had IPD. In strategy III, all pregnant women received SS, and those with T21 risk ≥1,000 had NIPT; then, women at high risk based on NIPT results had IPD. In strategy IV, all pregnant women received NIPT and those at high risk based on NIPT results had IPD. Finally, to assess the cost and effectiveness of DS screening, the total costs were calculated as the sum of screening and diagnosis as well as the direct and indirect economic burden during the average life cycle of DS patients.

### Results

A total of 22 of the 17,363 (1/789) pregnant women had DS, of which only one woman was over 35 years of age. SS detected 1,024 cases at high risk of T21 (≥1/270), 8 cases were

**Data Availability Statement:** We may not be able to make our underlying data set publicly available for legal or ethical reasons. But the data and materials used or analyzed during the current study are available from the corresponding author or the

Scientific Research and Education Department where the data access committee upon reasonable request. The email address for the Scientific Research and Education Department of our institution is zhfykjk@zhuhai.gov.cn.

**Funding:** Funding is provided by: Science and Technology planning projects of Zhuhai (Project No. ZH2202200063HJL).This project manager is the corresponding author of this manuscript.

**Competing interests:** The authors have declared that no competing interests exist.

true positive, with a positive predictive value of 0.78% and a detection rate of 36.4%. NIPT detected 27 cases at high risk of T21 ($Z \geq 3$) and 22 cases of DS, with a positive predictive value of 81.5% and a detection rate of 100%. Strategy I had the largest total cost of 65.54 million CNY, strategy II and III had similar total costs of 40 million CNY, and strategy IV had the lowest total cost of 14.91 million CNY. By comparison, the screening strategy with NIPT alone had the highest health economic value for DS.

## Conclusions

SS was greatly affected by nuchal translucency and the accuracy of gestational age measured by ultrasonography. Unstandardized ultrasonography was an important reason for the low DS detection rate with SS. The influence of interfering factors on NIPT was much lower than in SS. NIPT can be used as an alternative to SS and as a primary screening strategy of prenatal DS screening for secondary prevention and control of birth defects. NIPT greatly decreased the frequency of IPD and the miscarriages associated with IPD, saved the limited medical and health resources, and greatly increased DS detection rate. Therefore, NIPT has great social and economic benefits.

## Introduction

Down syndrome (DS) is caused by chromosome 21 trisomy (T21) and represents the most common genetic disorder associated with intellectual disability. The 200–300 genes on chromosome 21 and multiple epigenetic factors have been associated with the clinical manifestations of DS [1]. The incidence of DS is approximately 1/800 worldwide, 1/500 in the United States [2], and 14.7/10,000 in China [3]. DS has imposed a heavy economic and emotional burden on society and families. Secondary prevention and control of DS during gestation has become an important and urgent public health issue in all countries and regions.

Commonly used prenatal DS screening methods include serum screening (SS) and noninvasive prenatal testing (NIPT). As early as the 1980s, Cuckle et al. [4] used alpha-fetoprotein in maternal serum for prenatal DS screening. In 1999, Wald et al. [5] proposed a new algorithm for risk assessment in sequential screening for DS, which integrated pregnancy-associated plasma protein A (PAPP-A) and fetal nuchal translucency (NT) measured by ultrasonography in the first trimester with alpha-fetoprotein, unconjugated estriol (uE3), free beta-human chorionic gonadotropin (F β-hCG), and inhibin A in the second trimester. At a false positive rate of 1%, this screening strategy yielded a DS detection rate of up to 85%, while single screening in the first trimester or triple screening in the second trimester had DS detection rates of 72% and 46%, respectively. With the rapid development of next generation sequencing (NGS), maternal plasma DNA sequencing has revolutionized prenatal DS screening [6]. At the Chinese University of Hong Kong, Prof. Y M Dennis Lo and colleagues first reported the presence of cell-free DNA (cfDNA) in maternal plasma in 1997. In theory, analysis of fetal genetic materials [7] allows for noninvasive and risk-free genetic detection, such as direct detection with cfDNA in maternal plasma for evaluating the risk of fetal DS, rather than indirectly with maternal serum biomarkers. This opens a new era of NIPT.

The cost of SS and NIPT are very different; NIPT can be >10-fold more expensive than SS in some regions. The clinical application of SS and NIPT is dependent on price, affordability, and government subsidies. Various models have been established for cost-effectiveness

analysis (CEA) of NIPT in first-line screening and second-tier investigation [8–11]. In this study, we retrospectively analyzed the results of SS and NIPT and the pregnancy outcomes during follow-up of 17,363 pregnant women in Zhuhai City, China. We also performed a health economic evaluation of four screening strategies from a public health perspective. Our results provide a reference for the selection of screening strategies suitable for the situations in various regions.

## Materials and methods

### Subjects

17,363 case of pregnant women who participated voluntarily in the public health service program of prenatal testing for prevention and control of birth defects in Zhuhai from 2018 to 2019 received both SS and NIPT during the same gestation. For those at high risk ($\geq$1/270) and intermediate risk (1/1,000 to 1/270) for T21 in SS, amniocentesis was recommended as an invasive prenatal diagnosis (IPD) to analyze fetal karyotype. Those at high risk based on NIPT were recommended for amniocentesis or cordocentesis for fetal karyotype analysis. Those who received IPD were called at one week and one month after operation and one month after delivery to follow up the pregnancy outcomes. Alternatively, the Maternal and Child Health Information System of Zhuhai Health Bureau was queried for the outcomes. Those who did not receive IPD were called one month after the expected date of delivery or the information system was queried to follow up pregnancy outcomes. All examinations were approved by the ethics committee and signed by pregnant women with informed consent. All pregnant women do not include minors.

### Noninvasive prenatal testing

The cfDNA in maternal plasma was detected using the BIGSEQ 500 (MGI, China) high-throughput sequencing system, and fetal chromosome aneuploidy was analyzed via HALOS software. The NIFTY® reagent was purchased from BGI (China). Quality control of the data used in bioinformatics analysis were as follows: $\geq$3.5% fetal DNA per sample, 38–42% GC, $\geq$5.2 M original data, $\geq$3.5 M valid data, and a cutoff Z score of 3.

### Serum screening

The test was performed on an Auto DELFIA 1235 automatic time-resolved fluoroimmunoassay (TRFIA) system (PerkinElmer, USA) with TRFIA reagent. In first-trimester screening, risk was assessed using PAPP-A and F β-hCGin maternal peripheral venous serum, as well as fetal NT measured by ultrasonography. In second-trimester screening, risk assessment was based on alpha-fetoprotein, F β-hCG, and uE$_3$ in maternal serum. Risk assessment was done using the Lifecycle 4.0 software, with a cutoff value of 1/270 for high risk and a cutoff value of 1/1,000 for intermediate risk for T21.

### Chromosome karyotype analysis

Approximately 10 mL of amniotic fluid was drawn from pregnant women at G18–G24 W, centrifuged to remove the supernatant, and the precipitated amniotic fluid cells were cultured in sterile cell culture medium (GIBCO, USA). The 0.5–1 mL heparin-anticoagulated cord blood from pregnant women after G24 W was incubated in blood lymphocyte culture medium. At the metaphase of mitosis, chromosomes were subjected to conventional Giemsa-banded karyotyping at a resolution of 320–400 bands. Karyotype images were acquired using an Imager Z2 automatic chromosome karyotype analyzer (Zeiss, Germany), and the karyotype

description was based on the International System for Human Cytogenomic Nomenclature (ISCN 2016).

## Health economic evaluation

All test and follow-up results were subjected to retrospective analysis using the four following screening strategies. CEA was conducted based on the price and fee schedules of Zhuhai and the per capita gross domestic product (GDP) of Guangdong Province in 2019. The economic costs of DS mainly include DS screening, IPD diagnosis, IPD-related miscarriages, and the economic burden of the patient's family due to the disease. The economic burden of DS can be direct and indirect. Direct economic burden refers to the direct medical costs and direct non-medical costs derived from the treatment of DS, and the costs of development services and special education of DS patients. Indirect economic burden is the loss of labor productivity in DS patients and in their family members, which is estimated by the reduction of effective working time and productivity. Zeng et al. [12] at Central South University estimated that the economic burden of DS was about 1.1 million CNY per case based on the per capita GDP of 23,798 CNY in Hunan Province in 2010. According to the *China Statistical Yearbook 2020* [13], the per capita GDP of Guangdong Province in 2019 was 94,172 CNY. Therefore, the economic cost for each missed case of DS in Zhuhai was approximately 4.35 million CNY. In addition, according to the medical service price schedule from the Zhuhai Price Bureau, the cost of SS was 120 CNY/test, the cost of NIPT was 855 CNY/test, the cost of IPD (including abdominal paracentesis, ultrasound-guided abdominal paracentesis, and chromosome karyotype analysis) was 2,500 CNY/test, and the average cost of IPD-related miscarriages was 2,000 CNY/person. In the following formulas, A is the total number of pregnant women screened, B is the number of pregnant women at high risk of T21 in SS ($\geq$1/270), C is the number of pregnant women at intermediate risk of T21 in SS (1/1,000 to 1/270), D is number of pregnant women at high risk of T21 in NIPT, E is the number of missed cases of DS, and F is the number of IPD-related miscarriages. The values of D and E are dependent on the screening strategy. All costs calculated in this study were expressed in CNY.

Strategy I: All pregnant women received SS and those at high risk of T21 had IPD.

Cost = A × 120 + B × 2,500 + E × 4,350,000 + F × 2,000 (CNY)

Strategy II: All pregnant women received SS, those with T21 risk $\geq$1/270 had IPD, and those with 1/270 > T21 risk $\geq$1/1,000 received NIPT; women at high risk based on NIPT results had IPD.

Cost = A × 120 + C × 855 + (B + D) × 2,500 + E × 4,350,000 + F×2,000 (CNY)

Strategy III: All pregnant women received SS, and those with T21 risk $\geq$1/1,000 had NIPT; women at high risk based on NIPT results had IPD.

Cost = A × 120 + (B + C) × 855 + D × 2,500 + E × 4,350,000 + F × 2,000 (CNY)

Strategy IV: All pregnant women received NIPT and those at high risk for T21 had IPD.

Cost = A × 855 + D × 2,500 + E × 4,350,000 + F × 2,000 (CNY)

The flowchart of screening with the above four strategies is shown in Fig 1.

## Results

### Basic information

The data were collected from 19,465 pregnant women and 2,102 were lost to follow-up due to incomplete records of postpartum visits in the Maternal and Child Health Information System, change of telephone number, or unsuccessful telephone follow up, with a loss rate of 10.8%. Finally, a total of 17,363 pregnant women were enrolled, with a mean gestation period of G9–G30 W and a mean age of 28.9 ± 3.7 years, of which 796 (4.6%) were $\geq$35 years old. A total of

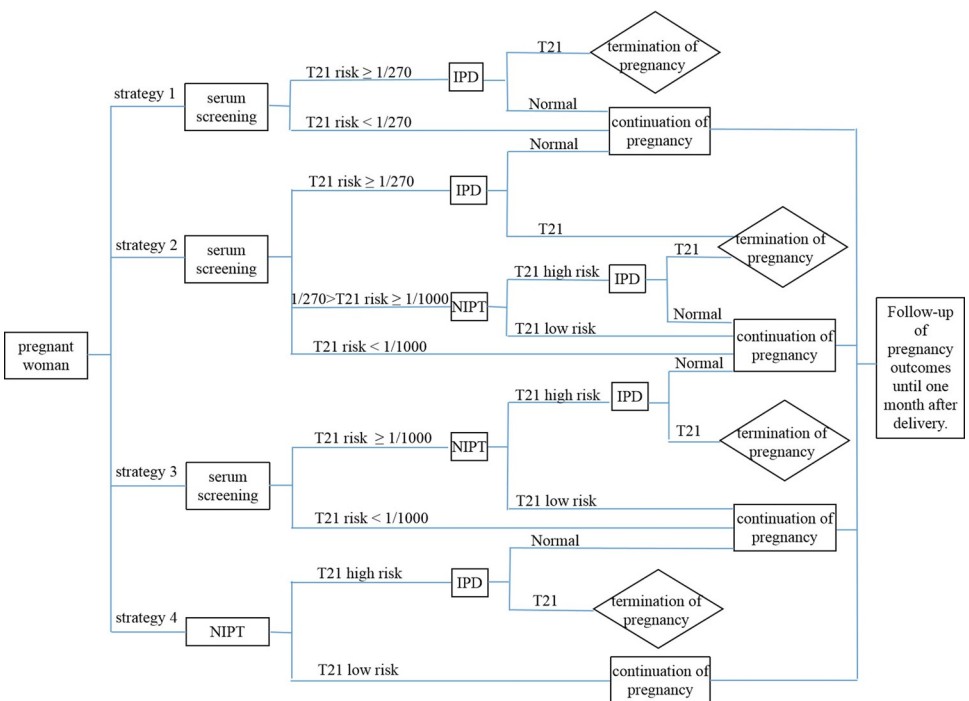

**Fig 1. Flowchart of four screening strategies.** NIPT, noninvasive prenatal testing; IPD, invasive prenatal diagnosis.

22 DS cases were detected, of which only one was 37 years old, and the remaining were <35, with an incidence of 1/789. One pregnant woman with IPD had a miscarriage within two weeks after the procedure.

## Serum screening

A total of 8,765 pregnant women participated in first-trimester screening, with a mean gestational age of 12.1 ± 0.7 W and a mean age of 29.1 ± 3.4 years. A total of 8,598 pregnant women participated in second-trimester screening, with a mean gestational age of 16.4 ± 1.2 W and a mean age of 28.7 ± 4.0 years. Of the 22 DS cases, 8 were detected in pregnant women with a T21 risk ≥1/270, giving a positive detection rate of 36.4% (8/22). Of the 14 missed DS cases, 10 were in the first trimester and 4 in the second trimester, and 6 had a T21 risk between 1/1,000 and 1/270. When the risk cutoff was expanded to 1/1,000, the additional 6 cases led to a positive detection rate of only 63.6% (14/22). Of 16 ture DS cases in first-trimester screening, for the 10 cases missed calculation without NT and with the serum indicators PAPP-A and F β-hCG identified 4 cases, and the detection rate was 62.5% (10/16) (Table 1) (Fisher exact test; P = 0.289).

## Noninvasive prenatal testing

NIPT screening showed 27 cases at high risk for T21, 22 confirmed cases, and 5 false positive cases. There were no reports of missed cases in the follow-up (Table 2). The results of SS screening and the fetal karyotype of IPD corresponding to the 27 high-risk cases in NIPT screening are shown in Table 3.

It can be seen from Tables 1 and 2 that although the NPV of NIPT is similar to that of SS, it is far superior to SS in terms of detection rate, IPD rate, PPV and FPR. as shown in Fig 2.

**Table 1. List of Serum screening results.**

| parameter | Number of pregnant women in first-trimester screening (%) | Number of pregnant women in second-trimester screening (%) | Total number of pregnant women screened (%) |
|---|---|---|---|
| T21 risk ≥ 1/270 | 361 (4.12) | 663 (7.71) | 1,024 (5.90) |
| T21 risk < 1/270 | 7,496 (85.52) | 6,379 (74.19) | 13,875 (79.91) |
| 1/1,000 ≤ T21 risk < 1/270 | 908 (10.36) | 1,556 (18.10) | 2,464 (14.19) |
| Detected cases[a] | 6 (0.06) | 2 (0.02) | 8 (0.05) |
| Undetected cases[a] | 10 (0.11) | 4 (0.05) | 14 (0.08) |
| FPR | 4.05% | 7.69% | 5.85% |
| PPV | 1.66% | 0.30% | 0.78% |
| NPV | 99.88% | 99.95% | 99.91% |
| Total | 8,765 (100.0) | 8,598 (100.0) | 17,363 (100.0) |

FPR, false positive rate; PPV, positive predictive value; NPV, negative predictive value.

[a] the number of detected cases is the number of DS cases with risk ≥1/270, and the number of missed cases is the number of DS cases with risk <1/270.

## Health economic evaluation of DS screening

A hypothetical retrospective analysis was performed on the data of four screening strategies from a public health perspective. In strategy IV, all pregnant women received NIPT and those at high risk had IPD, which was the optimal screening strategy with the lowest total cost. In strategies II and III, SS was performed for primary screening and NIPT for secondary screening, and those at high risk received IPD. Both strategies showed missed DS cases, and the total costs were approximately 2.7-fold that of strategy IV. In strategy I, DS detection with SS and IPD led to the highest number of missed cases, the highest total cost (50.63 million CNY, or 4.4-fold, more than strategy IV), and an additional cost of 2,916 CNY per case (Table 4).

## Discussion

SS is divided into first-trimester screening at G9–13$^{+6}$ W and second-trimester screening at 14–20$^{+6}$ W. The gestational age in the model was estimated by biparietal diameter of the fetal skull measured by ultrasonography, which was included in risk calculation in the first trimester. In another study, the use of gestational age estimated by ultrasonography significantly reduced the variation of indices within one week of pregnancy. These data suggest that routine

**Table 2. List of Noninvasive prenatal testing results.**

| parameter | Number of pregnant women screened for T21 (%) | | Total |
|---|---|---|---|
| | First trimester (≤13$^{+6}$ W) | Second and third trimesters (≥14 W) | Number of pregnant women screened for T21 (%) |
| Low risk | 4,483 (99.87) | 12,853 (99.84) | 17,336 (99.84) |
| High risk[a] | 6 (0.13) | 21 (0.16) | 27 (0.16) |
| Detected cases | 6 (0.13) | 16 (0.12) | 22 (0.13) |
| Undetected cases | 0 (0) | 0 (0) | 0 (0) |
| FPR | 0 | 0.11% | 0.03% |
| PPV | 100% | 76.2% | 81.5% |
| NPV | 100% | 100% | 100% |
| Total | 4,489 (100.0) | 12,874 (100.0) | 17,363 (100.0) |

FPR, false positive rate; PPV, positive predictive value; NPV, negative predictive value.

[a] Z ≥ 3 indicates high risk in NIPT.

**Table 3. Serum screening and fetal karyotype of cases at high risk in noninvasive prenatal testing.**

| Case number | Age (years) | Gestational weeks in NIPT | cfDNA (%) | Z value[a] | Gestational weeks in SS | T21 risk in SS[b] | Fetal karyotype |
|---|---|---|---|---|---|---|---|
| Case 1 | 30 | 18 | 9.64 | 17.78 | 16 | 1/97 | 47, XN[c], +21 |
| Case 2 | 26 | 17 | 9.14 | 11.17 | 16 | 1/58 | 46, XY, der(14;21)(q10;q10), +21 |
| Case 3 | 32 | 19 | 11.75 | 21.52 | 16 | 1/703 | 47, XX, +21 |
| Case 4 | 34 | 18 | 9.48 | 4.01 | 16 | 1/778 | 46, XX |
| Case 5 | 26 | 20 | 9.14 | 10.81 | 17 | 1/812 | 47, XY, +21 |
| Case 6 | 30 | 20 | 8.27 | 17.27 | 16 | 1/761 | 47, XX, +21 |
| Case 7 | 28 | 22 | 8.84 | 4.85 | 19 | 1/3,708 | 46, XY |
| Case 8 | 23 | 17 | 11.78 | 14.69 | 16 | 1/1,002 | 47, XX, +21 |
| Case 9 | 32 | 16 | 13.36 | 10.01 | 13 | 1/205 | 47, XY, +21 |
| Case 10 | 37 | 16 | 10.49 | 29.99 | 12 | 1/37 | 47, XN[c], +21 |
| Case 11 | 31 | 13 | 11.98 | 12.96 | 12 | 1/6 | 47, XX, +21 |
| Case 12 | 31 | 13 | 12.37 | 17.93 | 12 | 1/5 | 47, XN[c], +21 |
| Case 13 | 33 | 12 | 5.51 | 4.48 | 12 | 1/104 | 47, XX, +21 |
| Case 14 | 32 | 13 | 10.10 | 10.84 | 12 | 1/5 | 47, XX, +21 |
| Case 15 | 31 | 14 | 12.08 | 15.50 | 12 | 1/377 | 47, XX, +21 |
| Case 16 | 30 | 14 | 11.71 | 13.75 | 12 | 1/586 | 47, XY, +21 |
| Case 17 | 30 | 12 | 6.71 | 8.55 | 12 | 1/595 | 47, XX, +21 |
| Case 18 | 24 | 16 | 21.12 | 12.19 | 12 | 1/1,677 | 46, XX |
| Case 19 | 22 | 26 | 10.26 | 12.18 | 12 | 1/11,033 | 46, XY |
| Case 20 | 30 | 20 | 8.10 | 10.77 | 12 | 1/2,001 | 47, XY, +21 |
| Case 21 | 32 | 15 | 8.08 | 13.72 | 12 | 1/1,002 | 47, XX, +21 |
| Case 22 | 31 | 17 | 7.83 | 5.59 | 13 | 1/5,749 | 47, XY, +21 |
| Case 23 | 31 | 19 | 11.71 | 3.98 | 12 | 1/1,873 | 47, XY, +21[9]/46, XY[41] |
| Case 24 | 28 | 13 | 9.50 | 17.42 | 12 | 1/1,096 | 47, XX, +21 |
| Case 25 | 30 | 20 | 10.37 | 14.88 | 12 | 1/2,581 | 47, XX, +21 |
| Case 26 | 27 | 15 | 13.34 | 10.75 | 12 | 1/1,569 | 47, XY, +21 |
| Case 27 | 31 | 21 | 11.06 | 3.61 | 11 | 1/1,292 | 46, XX |

NIPT, noninvasive prenatal testing; SS, serum screening.

[a]$Z \geq 3$ indicates high risk in NIPT.

[b]risk $\geq 1/270$ indicates high risk in SS; and $1/1,000 \leq risk < 1/270$ indicates intermediate risk in SS.

[c] N indicates that invasive prenatal diagnosis was conducted in other institutions and fetal gender was not reported.

use of ultrasonography for gestational age estimation increased DS detection rate from 58% to 67% with a false positive rate of 5% [14]. However, there are multiple subjective factors such as detection technique, measurement method, clinical experience, and expertise of ultrasound technicians. The lack of standardized measurement in primary hospitals easily leads to large variation in the estimation of NT and gestational age, which in turn affects the accuracy of DS risk calculation. The use of NT in risk calculation for DS screening in the first trimester and the use of gestational age estimated by ultrasonography in the second trimesters led to a detection rate of SS much lower than theoretical value. In this study, double screening without NT detected 62.5% of DS cases in the first trimester, while screening with NT only detected 36.4% of DS cases. Although there was no significantly difference between the two detection rates (P = 0.289), it might be caused by insufficient samples. In this study, the <40% detection rate of triple screening in the second trimester was much lower than rates reported in various regions, which may be attributable to the great variation in gestational age measured by ultrasonography.

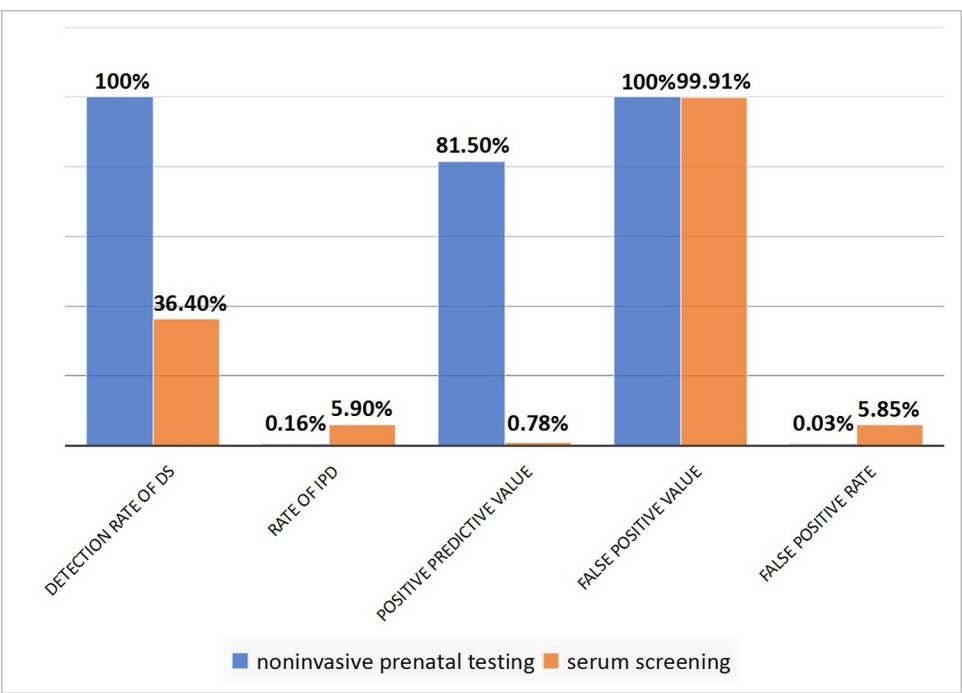

**Fig 2. Comparison of various indicators in the statistical results of NIPT and SS.**

Accumulating screening data have shown that DS screening with NIPT has unparalleled advantages over SS regarding PPV, FPV, and IPD. The data in this study also confirms this point. For example, we observed much higher PPV of NIPT (81.5%) than SS (0.78%). Moreover, SS is applicable only to pregnant women at G9–G20$^{+6}$ W, while NIPT is applicable at more than 21 weeks of gestation until last trimester. However, the NIPT technical specifications issued by the Health and Family Planning Commission of China in 2016 [15] recommended NIPT as a secondary strategy to SS. In the specification, NIPT is applicable to pregnant women whose SS shows a fetal aneuploidy risk between the high-risk cutoff value and 1/1,000 or who miss the optimal time for SS (≤G20$^{+6}$ W) but are required to perform risk assessment for trisomy 21, trisomy 18, and trisomy 13, or who have contraindications for IPD.

Because of religious beliefs, legal regulations, medical payment and other factors, different regions adopt different strategies when choosing state-sponsored DS screening. In the public health service program, cost is the most important factor preventing NIPT from replacing SS

**Table 4. Costs of DS screening with four screening strategies.**

| Screening strategy | Total number of pregnant women screened (A) | SS ≥ 1/270 (B) | SS 1/1,000–1/270 (C) | High risk in NIPT (D) | Missed DS cases in SS (E) | Number of IPD-related miscarriages (F) | Total costs (CNY) |
|---|---|---|---|---|---|---|---|
| I | 17,363 | 1,024 | – | – | 14 | 0 | 65,543,560 |
| II | 17,363 | 1,024 | 2,464 | 8[a] | 8 | 1 | 41,572,280 |
| III | 17,363 | 1,024 | 2,464 | 15[b] | 8 | 1 | 39,905,300 |
| IV | 17,363 | – | – | 27 | 0 | 0 | 14,912,865 |

SS, serum screening; DS, Down syndrome; NIPT, noninvasive prenatal testing; IPD, invasive prenatal diagnosis.

[a] denotes the number of cases at high risk in NIPT in C.

[b] denotes the number of cases at high risk in NIPT in B and C.

as a first-line prenatal DS screening strategy. Additionally, the cost-effectiveness ratio of the screening strategy is the most important consideration for local health administrations. Some countries effectively regulate the supply of NIPT on grounds of cost-effectiveness and reliability, there is disagreement regarding the implementation of NIPT in different nations [16], and health facilities worldwide have carried out various CEAs of SS and NIPT for DS screening. However, almost all these analyses were based on calculation with statistical models assuming SS as a first-line screening test and NIPT as a secondary strategy.

In 2014, the UK National Health Service (NHS) [8] used a pre-existing model to assess and compare the costs and outcomes of NIPT for DS. They found that NIPT as a contingent test would be cost neutral or cost saving compared with current DS screening if the cost of NIPT was <400 GBP and the screening risk cutoff was 1:150. NIPT as first-line testing would achieve more favorable outcomes but at a greater cost. Therefore, further research is needed to determine whether NIPT can be promoted as a first-line screening strategy in public health services. In the Netherlands, Beulen et al. [11] developed a decision-analytic model for CEA of prenatal DS screening in clinical practice. They found that the introduction of NIPT increased DS detection rate and decreased IPD, thereby decreasing IPD-related miscarriages and the cost of DS per case. Because of the high cost of NIPT as a new technology, it is not feasible to use NIPT as a primary screening test, but it should be used as an optional test for pregnancies at high risk for T21. In the United States in 2015, Evans et al. [17] performed a decision tree analysis and suggested that NIPT as a primary screening was not cost-effective, and the cost was lowest when it was used as a contingent strategy, especially with a risk cutoff of 1/1,000. Although the cost of a hybrid strategy was lower than that of NIPT as a primary strategy, the cost was higher than that of NIPT as a contingent strategy. The cost of NIPT in the United States was 1,017 USD/test.

In 2019, Zhang et al. [10] used a microsimulation decision-analytic model to conduct a sample survey of 45,605 pregnant women in British Columbia, Canada. They concluded that NIPT screening was more effective and more expensive, and NIPT at 200 USD or less was more cost-effective as a first-line screening strategy. Xu et al. [18] of Fudan University conducted a survey of physicians and experts from 25 medical facilities in Shanghai, Hunan Province, Zhejiang Province, and Shandong Province, China, as well as a literature search of relevant data. A decision-analytic model was established in a simulated cohort of 10,000 pregnant women using TreeAge Pro software, and a CEA of NIPT was performed from a societal perspective. The study found that NIPT was the most effective when used as a universal screening strategy because it detected more DS cases, and NIPT was the safest and most cost-effective as a contingent screening strategy. The cost evaluation of the study did not include direct non-medical costs and non-direct costs, which in fact bring the greatest economic burden of DS on society and families.

Our institute provides public health services such as prenatal screening of DS for more than 20,000 pregnant women each year in Zhuhai. Pregnant women have the option of either or both SS and NIPT tests, and the costs are covered by a special fund of Zhuhai Municipal Government and the maternity insurance in the social medical security fund. If the fetus is diagnosed with DS, before the fetus is born, the pregnant woman has the right to decide whether to terminate the pregnancy or give birth to the fetus. The data used for health economic evaluation in this study were based on a statistical analysis of the clinical test results and the postnatal follow-up outcomes of pregnant women who received both tests during gestation. Only 1 of the 22 DS cases in 17,363 pregnancies was identified in a woman with advanced age (>35 years), which was because most participants with advanced age received IPD instead of SS and NIPT. Based on the data in Zhuhai, it was estimated that SS as a single test or SS as a first-line test and NIPT as a secondary test had much higher total costs than NIPT as a first-line

screening strategy. With the rapid development of NGS and the decrease in sequencing costs, NIPT will become increasingly inexpensive, and the cost difference will be exacerbated. Moreover, the impact of subjective factors on NIPT was much lower than on SS. Therefore, secondary prevention and control of birth defects using NIPT instead of SS as a first-line test for prenatal DS screening can greatly reduce the frequency of IPD and IPD-related miscarriages, save the limited healthcare resources, greatly improve DS detection rate, and thus bring significant social and economic benefits. If the pregnancy woman with a DS fetus finally decides to have the baby, the parents still benefit from screening in being able to prepare for the birth of a child with disabilities, but it would affect the economic costs and benefits of the outcome.

## Acknowledgments

We thank Prof. Yuqiu Zhou at Zhuhai Institute of Medical Genetics for professional guidance on the research design, and the Prenatal Diagnostic Center of Zhuhai Maternal and Child Health Hospital for providing some of the data. The authors declare no conflict of interest.

## Author Contributions

**Data curation:** Wuyan Huang, Guoqing Wang.

**Funding acquisition:** Gefei Xiao.

**Investigation:** Liqing Hu.

**Methodology:** Huayu Luo.

**Project administration:** Gefei Xiao.

**Resources:** Yanling Zhao.

**Writing – original draft:** Gefei Xiao.

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
