## [Decision Letter · Decision Letter 0]

26 May 2021

PONE-D-21-11149

Health economic evaluation of noninvasive prenatal testing and serum screening for Down syndrome

PLOS ONE

Dear Dr. Xiao,

Thank you for submitting your manuscript to PLOS ONE. After careful consideration, we feel that it has merit but does not fully meet PLOS ONE’s publication criteria as it currently stands. Therefore, we invite you to submit a revised version of the manuscript that addresses the points raised during the review process.

We look forward to receiving your revised manuscript.

Kind regards,

Sherif A. Shazly, M.B.B.Ch

Academic Editor

PLOS ONE

Journal Requirements:

2. Thank you for including your ethics statement:  "All examinations were approved by the ethics committee of Zhuhai Center for Maternal and Child Healthcare and informed consent was obtained from pregnant women.The authors of the manuscript declare no conflicts of interest.".   

Please provide additional details regarding participant consent. In the ethics statement in the Methods and online submission information, please ensure that you have specified what type you obtained (for instance, written or verbal, and if verbal, how it was documented and witnessed). If your study included minors, state whether you obtained consent from parents or guardians. If the need for consent was waived by the ethics committee, please include this information.

3. Please amend the manuscript submission data (via Edit Submission) to include authors Guoqing Wang, Huayu Luo.

Reviewers' comments:

Reviewer #1: Dear authors,

Thanks for your submission. I just have some minor comments;

1- Please mention the full name of hormone uE3 before using the abbreviation at first mention in the document (line 106).

2- Titles of tables 1 and 2 need to be more informative.

3- In the discussion you mention that the DS risks calculated with or without the NT were significantly different in the first trimester (line 238), but i couldn't find such comparison in results section.

4- In the discussion you mention in line 253 that cfDNA is undetectable in maternal blood until 2 h after delivery. What is the significance of this information in relation to you discussion argument?

Finally the manuscript is generally acceptable and worth publishing after these minor rivisions

Thanks,

Reviewer #2: General Comments

The paper is interesting. The message is straight forward, and the results presented support the conclusion. The authors successfully demonstrate how NIPT can be used as a non-invasive primary step for the diagnosis Down Syndrome. They also provide evidence showing that it may be more cost effective and accurate than serum screening. However, the results are not discussed in depth and I see several other problems with the manuscript.

Firstly, there seems to be a lack of detail in the methods. While many of the details are ratified within the results, the Methods and Materials section at times is somewhat ambiguous, especially regarding the number of study participants and the included tests.

Secondly, the first page of the discussion, while interesting in context of the history of down syndrome screening, would probably fit better within the introduction. This first portion of the discussion makes the section very long and while it provides some context, it does not provide information that supports the presented results.

Thirdly, the paper would be well served by including graphical representation of the results, rather than including all results within tables. The one figure that was included was illegible.

Thirdly, while it is stated at line 72 “We also performed a health economic evaluation of four screening strategies from a public health perspective.” very little discussion is given to the possible economic impact or outcomes of the different screening strategies. In table 4 it is demonstrated that strategy 4 (where NIPT is the only screening method) is approximately 25,000,000 CNY cheaper than the next cheapest strategy and over 50,000,000 CNY cheaper than strategy 1, a factor of over 4 times cheaper. At line 260 it is stated “In the public health service program, cost is the most important factor preventing NIPT from replacing SS as a first-line prenatal DS screening strategy. Additionally, the cost-effectiveness ratio of the screening strategy is the most important consideration for local health administrations.” If this is the case, and strategy 4 is considerably cheaper with promising outcomes why is this not being considered or implemented? There is no critical discussion present on this. Furthermore, there seems to be no discussion over the advantages or disadvantages of using NIPT over SS as the primary screening method, apart from NIPT being less susceptible to subjective factors (which are not outlined).

Finally, at line 312 “Therefore, secondary prevention and control of birth defects using NIPT instead of SS as a first-line test for prenatal DS screening can greatly reduce the frequency of IPD and IPD-related miscarriages”. While I agree that this conclusion is correct based on the presented results, can this conclusion be made in regards to IPD-related miscarriage when a single miscarriage was recorded?

Specific Points

INTRODUCTION

1. Line 53: “200-300 genes” is there no better precise estimate?

2. Line 65: Is there any trimester separation for NIPT or is it as applicable through pregnancy?

METHODS AND MATERIALS

3. Line 81: Did all women undergo both SS and NIPT or was it a mixture? Not clear.

4. No where in the methods do the authors specify the number of women that were tested. I assume it was 17363 as mentioned in the introduction.

5. Line 83: Are there breakpoints which categorise high risk in the NIPT test like described for SS?

6. Line 84: Is there a reason why only amniocentesis was recommended for SS while amniocentesis or cordocentesis was recommended for NIPT?

7. Again, were both SS and NIPT performed on all the women or SS on one proportion and NIPT on another? Based on the results I assume SS and NIPT were both performed on all patients. Needs to be outlined better within the Methods.

8. Line 85: How many patients underwent IPD?

9. Line 157: Figure 1 is essentially unreadable. Needs to be remade.

RESULTS

NIPT

10. Line 190: Table 2: “FPV” not consistent with “FPR, False Positive Rate”

11. Line 192: “*Z ≥ 3 indicates high risk in NIPT” Z is not outlined in NIPT methods section as risk assessment is in the SS methods section (Line 106 - 108).

DISCUSSION

12. Line 285: Is this true? There is no reference for this statement. The NIH priced NIPT in the USA as $700-1000 per test in 2014. One would expect this to be cheaper 7 years later.

6. PLOS authors have the option to publish the peer review history of their article (what does this mean?). If published, this will include your full peer review and any attached files.

---

## [Author Response · Author response to Decision Letter 0]

27 Jun 2021

Journal Requirements:

1.Please ensure that your manuscript meets PLOS ONE's style requirements, including those for file naming. 

Response：Yes, the format of the manuscript has been edited to meet PLOS ONE's style requirements.

2.Thank you for including your ethics statement:  "All examinations were approved by the ethics committee of Zhuhai Center for Maternal and Child Healthcare and informed consent was obtained from pregnant women.The authors of the manuscript declare no conflicts of interest.".   

Please provide additional details regarding participant consent. In the ethics statement in the Methods and online submission information, please ensure that you have specified what type you obtained (for instance, written or verbal, and if verbal, how it was documented and witnessed). If your study included minors, state whether you obtained consent from parents or guardians. If the need for consent was waived by the ethics committee, please include this information.

Response：All pregnant women signed the informed consent form before the examination. The relevant content has been added in line 116.“All examinations were approved by the ethics committee and signed by pregnant women with informed consent. All pregnant women do not include minors.”

3.Please amend the manuscript submission data (via Edit Submission) to include authors Guoqing Wang, Huayu Luo.

Response：Data about Guoqing Wang and Huayu Luo has been amended via Edit Submission.

Reviewer #1: 

1-Response to comment: Please mention the full name of hormone uE3 before using the abbreviation at first mention in the document (line 106).

Response：the full name of hormone uE3 is “unconjugated estriol”. It has been added in line 133.

2-Response to comment: Titles of tables 1 and 2 need to be more informative.

Response：The titles of tables 1 and 2 have been amended as “List of Serum screening results”(Line 209) and “List of Noninvasive prenatal testing results”(Line 218).

3- Response to comment: In the discussion you mention that the DS risks calculated with or without the NT were significantly different in the first trimester (line 238), but i couldn't find such comparison in results section.

Response：It has been mentioned in line 206.“For the 10 cases missed in first-trimester screening, calculation without NT and with the serum indicators PAPP-A and F β-hCG identified 4 cases, and the detection rate was increased to 62.5% (10/16) (Table 1).”

4- Response to comment: In the discussion you mention in line 253 that cfDNA is undetectable in maternal blood until 2 h after delivery. What is the significance of this information in relation to you discussion argument?

Response：We mean that the applicable gestational week of NIPT is longer than that of SS. It has been modified to“Moreover, SS is applicable only to pregnant women at G9–G20+6 W, while NIPT is applicable at more than 21 weeks of gestation until last trimester”in line 272.

Reviewer #2: 

General Comments:

1- Response to comment: There seems to be a lack of detail in the methods. While many of the details are ratified within the results, the Methods and Materials section at times is somewhat ambiguous, especially regarding the number of study participants and the included tests.

Response：The number of study participants and the included tests have been added in lines 105-107:“17,363 case of pregnant women who participated voluntarily in the public health service program of prenatal testing for prevention and control of birth defects in Zhuhai from 2018 to 2019 received both SS and NIPT during the same gestation. ”In the "Noninvasive prenatal testing" and "Serum screening" section of Materials and Methods, there are detailed introductions about these two tests respectively.

2-Response to comment: The first page of the discussion, while interesting in context of the history of down syndrome screening, would probably fit better within the introduction. This first portion of the discussion makes the section very long and while it provides some context, it does not provide information that supports the presented results.

Response：Thank you for your reminder. In the revised manuscript, the general introduction about SS and NIPT on the first page of the discussion was moved to the introduction(line 78), and some sentences were deleted.

3- Response to comment: The paper would be well served by including graphical representation of the results, rather than including all results within tables. The one Fig that was included was illegible.

Response：This is a great suggestion. We have sorted out part of the data and shown it in Fig 2. The resolution of Fig 1 has been adjusted to 400dpi.

4-Response to comment: While it is stated at line 72 “We also performed a health economic evaluation of four screening strategies from a public health perspective.” very little discussion is given to the possible economic impact or outcomes of the different screening strategies. In table 4 it is demonstrated that strategy 4 (where NIPT is the only screening method) is approximately 25,000,000 CNY cheaper than the next cheapest strategy and over 50,000,000 CNY cheaper than strategy 1, a factor of over 4 times cheaper. At line 260 it is stated “In the public health service program, cost is the most important factor preventing NIPT from replacing SS as a first-line prenatal DS screening strategy. Additionally, the cost-effectiveness ratio of the screening strategy is the most important consideration for local health administrations.” If this is the case, and strategy 4 is considerably cheaper with promising outcomes why is this not being considered or implemented? There is no critical discussion present on this. 

Furthermore, there seems to be no discussion over the advantages or disadvantages of using NIPT over SS as the primary screening method, apart from NIPT being less susceptible to subjective factors (which are not outlined).

Response：The question you mentioned is the meaning of our article. Because almost all health economics evaluations were based on calculation with statistical models assuming SS as a first-line screening test and NIPT as a secondary strategy（Line 285）, there has been a lack of objective and actual data from clinical practice on large populations for reference to health administration departments. Because NIPT is currently available in the private sector in most country, the price of NIPT is expensive. This is also the main reason why the model data believes that SS is more advantageous than NIPT in cost-benefit analysis. However, the price of NIPT in Zhuhai City is much lower than that in other places due to the intervention of the government. So the total cost of NIPT as a first-line screening has been greatly reduced. But not every private sector is willing to reduce the price of NIPT to such a cheap price.

We have Added “The data in this study also confirms this point” in line 270. And in line 272 of the discussion, We also wrote that “Moreover, SS is applicable only to pregnant women at G9–G20+6 W, while NIPT is applicable at more than 21 weeks of gestation until last trimester”, which is also one of the advantages of NIPT. We also added Fig 2 in “Results” to show that NIPT has more advantages than SS.(Line 232)

5-Response to comment: Finally, at line 312 “Therefore, secondary prevention and control of birth defects using NIPT instead of SS as a first-line test for prenatal DS screening can greatly reduce the frequency of IPD and IPD-related miscarriages”. While I agree that this conclusion is correct based on the presented results, can this conclusion be made in regards to IPD-related miscarriage when a single miscarriage was recorded?

Response：The evidence of this conclusion is not because there was only one miscarriage, but the number of people with high risk of SS requiring IPD is much greater than that of NIPT, so the probability of miscarriage is greatly increased. This is a conclusion drawn from probability.

Specific Points:

Introduction

1. Response to comment: Line 53: “200-300 genes” is there no better precise estimate?

Response：There is no accurate number of genes associated with Down syndrome. Because the technology is constantly developing, new related genes are constantly being discovered, and the reported data in various places is sometimes not updated timely.

2. Response to comment: Line 65: Is there any trimester separation for NIPT or is it as applicable through pregnancy?

Response：NIPT is applicable at more than 12 weeks of gestation until last trimester. However, considering that high-risk pregnant women need to set aside enough time for interventional prenatal diagnosis, it is usually only recommended to have a maximum of 30 weeks of pregnancy.

Materials and methods:

3. Response to comment: Line 81: Did all women undergo both SS and NIPT or was it a mixture? Not clear.

Response：Yes. All pregnant women received both NIPT and SS during the same pregnancy. (Line 107) 

4. Response to comment: No where in the methods do the authors specify the number of women that were tested. I assume it was 17363 as mentioned in the introduction.

Response：Yes,you are right. It is mentioned in line 192 of the results.“Finally, a total of 17,363 pregnant women were enrolled, ” We have also added the number of women that were tested in “Subjects” of “Materials and methods”.(Line 105)

5.Response to comment: Line 83: Are there breakpoints which categorise high risk in the NIPT test like described for SS?

Response：Yes. It is mentioned in line 124 as“and a cutoff Z score of 3”.

6. Response to comment: Line 84: Is there a reason why only amniocentesis was recommended for SS while amniocentesis or cordocentesis was recommended for NIPT?

Response：The suitable gestational week for SS is 9-20+6 weeks, and the suitable gestational week for NIPT can be from 12 weeks to the third trimester. Generally, the most suitable gestational week for drawing amniotic fluid for karyotype analysis is 16-25 weeks. After 25 weeks, amniotic fluid cell culture is difficult to succeed, so cord blood is drawn for cell culture and karyotype analysis. Because the risk of miscarriage caused by amniocentesis is smaller than that of cord blood puncture, it is generally recommended to draw amniotic fluid for prenatal diagnosis for pregnant women at high risk of SS.

 However, some pregnant women may have been taking NIPT for more than 25 weeks, and it is no longer suitable to draw amniotic fluid for karyotyping, so it is recommended to draw cord blood. Although in the third trimester of DS diagnosis, in addition to the method of karyotyping, DNA of amniotic fluid cells can also be extracted to analyze chromosome data, but because of the high cost of testing, karyotyping of cord blood cells is still the first choice.

7. Response to comment: Again, were both SS and NIPT performed on all the women or SS on one proportion and NIPT on another? Based on the results I assume SS and NIPT were both performed on all patients. Needs to be outlined better within the Methods.

Response：Yes,you are right. All pregnant women received both NIPT and SS during the same pregnancy.(Line 107)

8. Response to comment: Line 85: How many patients underwent IPD?

Response：Regrettably, because some pregnant women did IPD not in our hospital, we can only see the final pregnancy outcome through the follow-up of the information system,and it,s not certain whether the pregnant woman had IPD. So there is no exact data on the total IPD.

9. Response to comment: Line 157: Fig 1 is essentially unreadable. Needs to be remade.

Response：Well, We have remade Fig 1. The resolution of Fig 1 has been adjusted to 400dpi.

Results

10. Response to comment: Line 190: Table 2: “FPV” not consistent with “FPR, False Positive Rate”

Response：We are very sorry for our negligence of it. It should be "FPR". It has been corrected now.(Line 218: Table 2)

11. Response to comment: Line 192: “*Z ≥ 3 indicates high risk in NIPT” Z is not outlined in NIPT methods section as risk assessment is in the SS methods section (Line 106 - 108).

Response：It has been described as“≥3.5% fetal DNA per sample, 38–42% GC, ≥5.2 M original data, ≥3.5 M valid data, and a cutoff Z score of 3”. (line 124)

Discussion

12. Response to comment: Line 285: Is this true? There is no reference for this statement. The NIH priced NIPT in the USA as $700-1000 per test in 2014. One would expect this to be cheaper 7 years later.

Response：Yes, the price of NIPT in the United States in 2021 may be cheaper. However, this sentence in this manuscript is derived from the results section of the abstract of Reference 17 “Utilizing a primary cffDNA screening strategy,the cost per patient was 1017 US$.” We have already stated the year of publication in the manuscript is 2015.(Line 300) We regret that we did not find a newer price of NIPT in US.

18.Response to comment: PLOS authors have the option to publish the peer review history of their article (what does this mean?). If published, this will include your full peer review and any attached files.

Response：I choose no.

---

## [Decision Letter · Decision Letter 1]

16 Nov 2021

PONE-D-21-11149R1Health economic evaluation of noninvasive prenatal testing and serum screening for Down syndromePLOS ONE

Dear Dr. Xiao,

Thank you for submitting your manuscript to PLOS ONE. After careful consideration, we feel that it has merit but does not fully meet PLOS ONE’s publication criteria as it currently stands. Therefore, we invite you to submit a revised version of the manuscript that addresses the points raised during the review process. Please address reviewer 1's reiterated question about your statistical analysis.

We look forward to receiving your revised manuscript.

Kind regards,

Yann Benetreau, PhD

Senior Editor, *PLOS ONE*

Journal Requirements:

Reviewers' comments:

Reviewer's Responses to Questions

**Comments to the Author**

1. If the authors have adequately addressed your comments raised in a previous round of review and you feel that this manuscript is now acceptable for publication, you may indicate that here to bypass the “Comments to the Author” section, enter your conflict of interest statement in the “Confidential to Editor” section, and submit your "Accept" recommendation.

Reviewer #1: (No Response)

Reviewer #2: All comments have been addressed

2. Is the manuscript technically sound, and do the data support the conclusions?

Reviewer #1: Yes

Reviewer #2: Yes

3. Has the statistical analysis been performed appropriately and rigorously? 

Reviewer #1: I Don't Know

Reviewer #2: I Don't Know

4. Have the authors made all data underlying the findings in their manuscript fully available?

Reviewer #1: Yes

Reviewer #2: No

5. Is the manuscript presented in an intelligible fashion and written in standard English?

Reviewer #1: Yes

Reviewer #2: Yes

6. Review Comments to the Author

Reviewer #1: Dear authors,

Thanks for your response. I just have some minor comments.

1- the first mentioning of uE3 is now moved to line 81 (this part was not included in your original submission), hence you should the full name of this abbreviation accordingly.

2- Your response to my 3rd comment is inadequate : (It has been mentioned in line 206.“For the 10 cases missed in first-trimester screening, calculation without NT and with the serum indicators PAPP-A and F β-hCG identified 4 cases, and the detection rate was increased to 62.5% (10/16) (Table 1).”). This doesn't necessarily imply significance. There has to be a p-value of at least <0.05 based on a suitable statistical test to conclude significant difference.

Reviewer #2: The authors have taken time to address each of my comments to a degree which I believe to be satisfactory.

7. PLOS authors have the option to publish the peer review history of their article (what does this mean?). If published, this will include your full peer review and any attached files.

Reviewer #1: No

Reviewer #2: No

---

## [Author Response · Author response to Decision Letter 1]

15 Dec 2021

1- the first mentioning of uE3 is now moved to line 81 (this part was not included in your original submission), hence you should the full name of this abbreviation accordingly.

Response：the full name of hormone uE3 is “unconjugated estriol”. It has been added in line 83. 

2- Your response to my 3rd comment is inadequate : (It has been mentioned in line 206.“For the 10 cases missed in first-trimester screening, calculation without NT and with the serum indicators PAPP-A and F β-hCG identified 4 cases, and the detection rate was increased to 62.5% (10/16) (Table 1).”). This doesn't necessarily imply significance. There has to be a p-value of at least <0.05 based on a suitable statistical test to conclude significant difference.

Response：We added statistical test result in line 205, and analyzed the reason in line 262 of discussion.

3-Have the authors made all data underlying the findings in their manuscript fully available?

Response：We have added “Availability of data and materials” in line 340.

---

## [Decision Letter · Decision Letter 2]

21 Feb 2022

PONE-D-21-11149R2

Health economic evaluation of noninvasive prenatal testing and serum screening for Down syndrome

PLOS ONE

Dear Dr. Xiao,

Thank you for submitting your manuscript to PLOS ONE. After careful consideration, we feel that it has merit but does not fully meet PLOS ONE’s publication criteria as it currently stands. Therefore, we invite you to submit a revised version of the manuscript that addresses the points raised during the review process.

Both reviewers are now happy with the technical aspects of the paper. I am not the original editor responsible for this paper, so I apologize, but I am going to request two small additional considerations. First, this is a topic where there are ethical issues involved, concerning the rights of those living with disabilities. I request the authors to refer briefly to this literature in the Introduction (a couple of sentences plus references would suffice).

Second, it would be helpful in the Discussion section to note that the results for China may not transfer readily to other countries. In some other countries, late detection of Down's syndrome may be too late for termination of the pregnancy to be legally permitted. Even where it is legally permitted, not all parents may exercise the choice to terminate the pregnancy. I don't know if this is an option in China (would Zhuhai Municipality permit the parents to continue on with the pregnancy?). In these circumstances the parents still benefit from screening in being able to prepare for the birth of a differently-abled child, but it would affect the economic costs and benefits of the outcome. Again, a couple of sentences qualifying the findings would suffice.

We look forward to receiving your revised manuscript.

Kind regards,

Susan Horton

Academic Editor

PLOS ONE

Journal Requirements:

Reviewers' comments:

Reviewer's Responses to Questions

**Comments to the Author**

1. If the authors have adequately addressed your comments raised in a previous round of review and you feel that this manuscript is now acceptable for publication, you may indicate that here to bypass the “Comments to the Author” section, enter your conflict of interest statement in the “Confidential to Editor” section, and submit your "Accept" recommendation.

Reviewer #1: All comments have been addressed

2. Is the manuscript technically sound, and do the data support the conclusions?

Reviewer #1: Yes

3. Has the statistical analysis been performed appropriately and rigorously? 

Reviewer #1: I Don't Know

4. Have the authors made all data underlying the findings in their manuscript fully available?

Reviewer #1: No

5. Is the manuscript presented in an intelligible fashion and written in standard English?

Reviewer #1: Yes

6. Review Comments to the Author

Reviewer #1: (No Response)

7. PLOS authors have the option to publish the peer review history of their article (what does this mean?). If published, this will include your full peer review and any attached files.

Reviewer #1: No

---

## [Author Response · Author response to Decision Letter 2]

25 Mar 2022

Response:

1- This is a topic where there are ethical issues involved, concerning the rights of those living with disabilities. I request the authors to refer briefly to this literature in the Introduction (a couple of sentences plus references would suffice).

Response：Because of religious beliefs, legal regulations, medical payment and other factors, different regions adopt different strategies when choosing state-sponsored DS screening. In China, termination of pregnancy is legally permitted when the fetus is diagnosed to be disabled before birth. It has been description in line 278. 

2- It would be helpful in the Discussion section to note that the results for China may not transfer readily to other countries. In some other countries, late detection of Down's syndrome may be too late for termination of the pregnancy to be legally permitted. Even where it is legally permitted, not all parents may exercise the choice to terminate the pregnancy. I don't know if this is an option in China (would Zhuhai Municipality permit the parents to continue on with the pregnancy?). In these circumstances the parents still benefit from screening in being able to prepare for the birth of a differently-abled child, but it would affect the economic costs and benefits of the outcome. Again, a couple of sentences qualifying the findings would suffice.

Response：If the fetus is diagnosed with DS, before the fetus is born, the pregnant woman has the right to decide whether to terminate the pregnancy or give birth to the fetus.We have added your comments to lines 283, 324 and 339.

---

## [Editor Report · Decision Letter 3]

28 Mar 2022

Health economic evaluation of noninvasive prenatal testing and serum screening for Down syndrome

PONE-D-21-11149R3

Dear Dr. Xiao,

We’re pleased to inform you that your manuscript has been judged scientifically suitable for publication and will be formally accepted for publication once it meets all outstanding technical requirements.

Kind regards,

Susan Horton

Academic Editor

PLOS ONE

Additional Editor Comments (optional):

please accept 2 small edits to the text (for English-language speakers): in line 337, delete the word "born" (it is ok to say "have the baby") and instead of "disability child" in line 338 please say "child with disabilities": in the literature on disability, the convention now is to put the person first, followed by the term describing their abilities.
---

## [Editor Report · Acceptance letter]

5 Apr 2022

PONE-D-21-11149R3 

Health economic evaluation of noninvasive prenatal testing and serum screening for Down syndrome 

Dear Dr. Xiao:

I'm pleased to inform you that your manuscript has been deemed suitable for publication in PLOS ONE. Congratulations! Your manuscript is now with our production department. 

Kind regards, 

on behalf of

Dr. Susan Horton 

Academic Editor

PLOS ONE